# Reduced Expression of Very-Long-Chain Acyl-CoA Synthetases *SLC27A4* and *SLC27A6* in the Glioblastoma Tumor Compared to the Peritumoral Area

**DOI:** 10.3390/brainsci13050771

**Published:** 2023-05-07

**Authors:** Jan Korbecki, Klaudyna Kojder, Dariusz Jeżewski, Donata Simińska, Patrycja Tomasiak, Maciej Tarnowski, Dariusz Chlubek, Irena Baranowska-Bosiacka

**Affiliations:** 1Department of Biochemistry and Medical Chemistry, Pomeranian Medical University in Szczecin, Powstańców Wlkp. 72, 70-111 Szczecin, Poland; jan.korbecki@onet.eu (J.K.); donata.siminska@pum.edu.pl (D.S.); dchlubek@pum.edu.pl (D.C.); 2Department of Anatomy and Histology, Collegium Medicum, University of Zielona Góra, Zyty 28 St., 65-046 Zielona Góra, Poland; 3Department of Anaesthesiology and Intensive Care, Pomeranian Medical University in Szczecin, Unii Lubelskiej 1, 71-281 Szczecin, Poland; klaudynakojder@gmail.com; 4Department of Neurosurgery and Pediatric Neurosurgery, Pomeranian Medical University in Szczecin, Unii Lubelskiej 1, 71-252 Szczecin, Poland; djezewski@wp.pl; 5Department of Applied Neurocognitivistics, Pomeranian Medical University in Szczecin, Unii Lubelska 1, 71-252 Szczecin, Poland; 6Department of Physiology, Pomeranian Medical University in Szczecin, Powstańców Wlkp. 72, 70-111 Szczecin, Poland; patrycja.tomasiak@pum.edu.pl (P.T.); maciejt@pum.edu.pl (M.T.)

**Keywords:** glioblastoma, fatty acid, solute carrier family 27, fatty acid transport protein, brain tumor

## Abstract

This study aimed to analyze solute carrier family 27 (SLC27) in glioblastoma tumors. The investigation of these proteins will provide insight into how and to what extent fatty acids are taken up from the blood in glioblastoma tumors, as well as the subsequent fate of the up-taken fatty acids. Tumor samples were collected from a total of 28 patients and analyzed using quantitative real-time polymerase chain reaction (qRT–PCR). The study also sought to explore the relationship between SLC27 expression and patient characteristics (age, height, weight, body mass index (BMI), and smoking history), as well as the expression levels of enzymes responsible for fatty acid synthesis. The expression of SLC27A4 and SLC27A6 was lower in glioblastoma tumors compared to the peritumoral area. Men had a lower expression of SLC27A5. Notably, a positive correlation was observed between the expression of SLC27A4, SLC27A5, and SLC27A6 and smoking history in women, whereas men exhibited a negative correlation between these SLC27s and BMI. The expression of SLC27A1 and SLC27A3 was positively correlated with the expression of ELOVL6. In comparison to healthy brain tissue, glioblastoma tumors take up fewer fatty acids. The metabolism of fatty acids in glioblastoma is dependent on factors such as obesity and smoking.

## 1. Introduction

Glioblastoma is the most aggressive form of glioma, categorized as grade 4 [1]. Its annual incidence is approximately four cases per 100,000 people [2,3]. Despite current treatment modalities, which include surgical resection, radiation therapy, and chemotherapy with temozolomide (TMZ) [4,5], this brain tumor has an exceedingly dismal prognosis. The median survival rate of patients with this malignancy does not surpass 1.5 years [2], and only a small fraction of patients, roughly 7%, survive beyond five years following diagnosis [2]. Given the extremely unfavorable prognosis for patients with glioblastoma, extensive research is underway to better understand the tumor mechanisms of this disease and develop novel therapeutic approaches. One of the areas of investigation in glioblastoma research pertains to fatty acid metabolism within the tumor.

Fatty acids represent a fundamental component of both cell and intracellular membranes, as well as a crucial source of energy and a precursor to the production of lipid mediators. The synthesis of fatty acids can occur de novo or, in the case of essential polyunsaturated fatty acids (PUFA), through the elongation and desaturation of other PUFAs [6]. The uptake of fatty acids by cells is facilitated by a variety of transporters, including the cluster of differentiation 36 (CD36) [7] and solute carrier family 27 (SLC27) [8]. For fatty acids to be utilized in cellular metabolism, they must first be activated by conversion to a fatty acyl coenzyme A (CoA) through the catalytic activity of acyl-CoA synthetases [8]. The SLC27A family comprises six members, denoted as SLC27A1 to SLC27A6 [8]. Previously, these proteins were referred to as very-long-chain acyl-CoA synthetases (ACSVL) or fatty acid transport proteins (FATP). The primary function of these enzymes is to catalyze the conversion of unesterified fatty acids to fatty acyl-CoA, while also facilitating the transport of fatty acids across the cell membrane. It is important to note that these enzymes must be localized in the cell membrane to carry out their transport function directly. Another important feature of acyl-CoA synthetases is the channeling of fatty acids [8], whereby a specific fatty acid is directed towards a particular metabolic pathway by a corresponding acyl-CoA synthetase, dependent on protein–protein interactions and subcellular localization of the synthetase [9].

SLC27A1/ACSVL5/FATP1 exhibits activity towards saturated fatty acids (SFA), particularly palmitic acid C16:0, as well as a much higher activity towards lignoceric acid C24:0 [10,11,12]. It can also activate arachidonic acid C20:4n-6 [13]. SLC27A1 is responsible for fatty acid transport [14]. SLC27A1 is located in the endoplasmic reticulum [15] and plays a role in triglyceride and ceramide synthesis [9,16,17,18]. It also directs fatty acids towards β-oxidation [18,19]. There is high expression of SLC27A1 in the brain [20,21] where it participates in the transport of fatty acids across the blood–brain barrier (BBB) in conjunction with SLC27A4 [22], with a particular role in the transport of docosahexaenoic acid (DHA) C22:6n-3 [23,24].

SLC27A2/ACSVL1/FATP2 is an enzyme that activates SFAs, including palmitic acid C16:0 and lignoceric acid C24:0 [25], as well as arachidonic acid C20:4n-6 [13]. SLC27A2 in peroxisomes plays a role in the β-oxidation of very-long-chain fatty acids [26] and contributes to the synthesis of glycerophospholipids, specifically in the incorporation of arachidonic acid C20:4n-6 and DHA C22:6n-3 [27]. Additionally, SLC27A2 is involved in ceramide synthesis [9] and is responsible for fatty acid uptake [13,28]. In the brain, SLC27A2 is expressed at very low levels [21,29,30].

SLC27A3/ACSVL3/FATP3 is an enzyme that activates SFA, namely stearic acid C18:0, behenic acid C22:0, and lignoceric acid C24:0 [31,32]. SLC27A3 participates in the synthesis of ceramides, as demonstrated by experiments on U87 MG cells [32]. Notably, SLC27A3 does not play a role in fatty acid uptake [13]. Furthermore, SLC27A3 is involved in the tumorigenesis of glioblastomas, particularly in the functioning of glioblastoma stem cells [33]. However, expression of SLC27A3 in the brain is very low [21,31].

SLC27A4/ACSVL4/FATP4 demonstrates activity towards palmitic acid C16:0 and significantly greater activity towards lignoceric acid C24:0 [12,34]. It also activates arachidonic acid C20:4n-6 [13]. SLC27A4 is responsible for fatty acid uptake [13,28,35]. This protein participates in the production of triglycerides [17,36], cholesterol esters [36], and ceramide [9]. SLC27A4 is highly expressed in the brain [21,34]. In the brain, SLC27A1 and SLC27A4 work together in the transport of fatty acids through the BBB [22].

SLC27A5/ACSVL6/FATP5 shows activity towards very-long-chain SFA, particularly stearic acid C18:0, and cerotic acid C26:0 [37]. Additionally, this enzyme displays bile acid-CoA ligase activity [38]. SLC27A5 is localized in the cell membrane and is responsible for fatty acid uptake, resulting in the production of triglycerides, glycerophospholipids, and cholesteryl esters [39]. The expression of SLC27A5 is very low in the brain [21,39].

SLC27A6/ACSVL2/FATP6 is a cardiac-specific isoform [40]. SLC27A6 activates arachidonic acid C20:4n-6 and exhibits weak very-long-chain acyl-CoA synthetase activity towards lignoceric acid C24:0 [13]. This protein preferentially transports palmitic acid C16:0 [40]. The expression of SLC27A6 is very low in the brain [21].

Recent research has shown that long-chain and very-long-chain fatty acids can initiate signal transduction by interacting with receptors or transporters. One example is CD36, which can activate a number of signaling pathways, including cAMP, Ca^2+^, Src (which activates phospholipase C (PLC), liver kinase B1 (LKB1), and insulin receptor β (IRβ)), ERK MAPK, vascular endothelial growth factor receptor 2 (VEGFR2), and peroxisome proliferator-activated receptor δ (PPARδ) [41]. Furthermore, long-chain and very-long-chain fatty acids can function as ligands for G protein-coupled receptors, including free fatty acid receptor (FFAR)1/G protein-coupled receptor (GPR)40 and FFAR4/GPR120 [41,42]. FFAR1/GPR40 is associated with G_q_, G_i_, and G_s_, while FFAR4/GPR120 only with G_q_. However, according to data from the GEPIA portal, these receptors do not appear to play significant roles in glioblastoma cancer processes. Specifically, the expression levels of FFAR1/GPR40 and FFAR4/GPR120 do not differ significantly between GBM tumors and healthy brain tissue, nor are they significantly associated with patient prognosis [43].

Acyl-CoA synthetases can trigger signal transduction upon activation by fatty acids, utilizing unesterified fatty acids, free coenzyme A (CoASH), and ATP as substrates, and producing fatty acyl-CoA and AMP as products [44,45]. AMP functions as a second messenger and is capable of activating AMP-activated protein kinase (AMPK), thereby participating in signal transduction [45].

In our previous investigations, we examined the expression and function of desaturases [46] and elongases [47] in glioblastoma. The findings of our research provided novel insights into the fatty acid biosynthesis within glioblastoma tumors. Nevertheless, the lipid metabolism in this malignancy remains insufficiently elucidated. The objective of this study is to examine the expression of SLC27 in glioblastoma tumors and to investigate whether patient characteristics are linked to the expression of these genes. Furthermore, the study aims to explore the correlation between the expression of SLC27 and the essential enzymes responsible for fatty acid synthesis in glioblastoma tumors.

## 2. Materials and Methods

### 2.1. Patient Samples

In 2014, the Department of Biochemistry and the Department of Neurosurgery and Pediatric Neurosurgery at Pomeranian Medical University in Szczecin initiated a research project that aimed to investigate purinergic receptors in glioblastoma tumors. Samples of glioblastoma tumors were obtained from consenting patients, and the project received ethical approval from the local bioethical commission (KB-0012/96/14) and was conducted in accordance with the Declaration of Helsinki. The research project was subsequently expanded to include the analysis of enzymes involved in lipid metabolism, which was also approved by the local bioethical commission (KB-0012/96/14-A-3).

The study analyzed SLC27 genes in 28 glioblastoma tumors obtained from 16 male and 12 female patients at the Department of Neurosurgery and Pediatric Neurosurgery of the Pomeranian Medical University in Szczecin, Poland (Table 1). Patients were diagnosed with brain tumors using magnetic resonance imaging (MRI) or computed tomography (CT) scans. Upon detection of a brain tumor, patients were eligible for surgical tumor removal. Surgery was performed under standard general anesthesia with endotracheal intubation. Craniotomy and tumor resection were performed according to the classical method, which involved bone removal and dura incision, tumor visualization, resection, biopsy for histopathological examination, closure of the dura, bone restoration in some patients, and the closure of subcutaneous tissue and skin in others. A computer station of a neuronavigation device was used during the surgical procedure to estimate the position of surgical instruments in relation to the glioblastoma tumor with a precision of 2–3 mm. After surgical tumor removal, a histopathological examination was performed to confirm the glioblastoma diagnosis. Patients included in the study were over 18 years old and had been diagnosed with glioblastoma.

Tumor samples were collected from two regions of the glioblastoma tumor (Figure 1) [48,49]: the non-enhancing tumor core located in the central part of the glioblastoma tumor and the enhancing tumor region, which usually surrounds the tumor core. A sample was also taken from the peritumoral area, which is a commonly used control in glioblastoma research that is compared to the glioblastoma tumor [50].

### 2.2. Quantitative Real-Time Polymerase Chain Reaction (qRT–PCR)

Two-step qRT–PCR was used to analyze the expression of *SLC27* genes in the U-87 MG glioblastoma cell line and patient samples. RNA was extracted from clinical specimens using the RNeasy Lipid Tissue Mini Kit (Qiagen, Hilden, Germany) and from U-87 MG cells using the RNeasy Mini Kit (Qiagen). The First Strand cDNA synthesis kit and oligo-dT primers (Fermentas, Waltham, MA, USA) were used for cDNA synthesis. Quantitative analysis of selected mRNAs was performed using an ABI 7500 Fast instrument with Power SYBR Green PCR Master Mix reagent (Applied Biosystems, Waltham, MA, USA) with the following PCR protocol: 95 °C for 15 s, 40 cycles at 95 °C for 15 s, and 60 °C for 1 min. Ct values were used for subsequent analysis, and *glyceraldehyde-3-phosphate dehydrogenase* (*GAPDH*) was used as the endogenous control [51]. The results were calculated as the fold difference (2^dCt) and subjected to statistical analysis. The data were presented as the absolute expression of the tumor tissue. The following primer pairs were used:
hSLC27A1FGGGGCAGTGTCTCATCTATGG;hSLC27A1RCCGATGTACTGAACCACCGT;hSLC27A2FTTTCCGCCATCTACACAGTCC;hSLC27A2RCGTAGGTGAGAGTCTCGTCG;hSLC27A3FCCCTGCTGGAATTAGCGATTT;hSLC27A3RGGGCGAGGTAGATCACATCTT;hSLC27A4FGGACCCAGGTGGGATTCTC;hSLC27A4RCGCGCCTGATGGTCTTGAT;hSLC27A5FTGGAGGAGATCCTTCCCAAGC;hSLC27A5RTGGTCCCCGAGGTATAGATGAA;hSLC27A6FCTTCTGTCATGGCTAACAGTTCT;hSLC27A6RAGGTTTCCGAGGTTGTCTTTTG.

In this study, we did not obtain results for *SLC27A2*, characterized by very low expression in the brain [21,29,30]. GEPIA analyses also show that the level of *SLC27A2* expression is about 200 times lower in glioblastoma than in other SLC27 genes [43].

### 2.3. Statistical Methods

The expression levels of SLC27 members were normalized to the reference gene *GAPDH*. Statistical significance was analyzed using Statistica v.13 software. Results with *p* ≤ 0.05 were considered statistically significant. The Shapiro–Wilk test indicated that the results did not follow a normal distribution. Therefore, non-parametric tests were used. Wilcoxon’s signed-rank tests were used to analyze differences in *SLC27* expression between different regions and under different culture conditions. Spearman rank correlation coefficients were used to analyze correlations between *SLC27* genes, patient characteristics, and expression levels of enzymes involved in the fatty acid synthesis.

## 3. Results

### 3.1. SLC27A4 and SLC27A6 Expression in the Glioblastoma Tumor Was Lower Than in the Peritumoral Area

Changes in the expression of individual SLC27 genes may contribute to alterations in the metabolism of fatty acids in glioblastoma tumors. Therefore, in the first stage of the study, the expression of *SLC27A* genes was examined in two regions of the glioblastoma: the enhancing tumor region and the tumor core. The expression of selected genes was also analyzed in the peritumoral area, which is commonly considered as a control to which the results of the glioblastoma tumor analysis can be compared [50].

The expression levels of *SLC27A1*, *SLC27A3*, and *SLC27A5* did not differ between the two analyzed regions of the glioblastoma tumor and the peritumoral area in all patients (Figure 2). However, in the enhancing tumor region and the tumor core, the expression of *SLC27A4* was significantly lower compared to the peritumoral area (*p* = 0.0007 and *p* = 0.004, respectively). Similarly, the expression of *SLC27A6* was also significantly lower in the enhancing tumor region and the tumor core compared to the peritumoral area (*p* = 0.014 and *p* = 0.0019, respectively).

### 3.2. When Comparing Tumors between Sexes, the Expression of Different SLC27As Differed in Glioblastoma Tumors

Lipid metabolism is dependent on sex, particularly on sex hormones. Studies on breast cancer have shown that estrogen receptor-β (ER-β) increases the expression of *SLC27A1* [52]. Estrogen receptors also influence the expression of long-chain acyl-CoA synthetases, other enzymes responsible for the activation of fatty acids. Specifically, estrogen receptor-α (ER-α) decreases the expression of acyl-CoA synthetase long-chain family member 4 (*ACSL4)* [53]. Due to the possibility of differences in the expression of the investigated SLC27 genes based on the sex of the patients, the expression results of SLC27A representatives were analyzed by sex and compared to each other.

The expression of *SLC27A1* was reduced in the tumor core of female glioblastoma patients compared to the enhancing tumor region (*p* = 0.03) and peritumoral area (*p* = 0.0002). Additionally, the expression of *SLC27A1* in the tumor core of female glioblastoma patients was lower than in male patients (*p* = 0.000003).

In the studied regions of glioblastoma tumors, the expression of *SLC27A3* did not differ from the peritumoral area in males and females. However, in the enhancing tumor region (*p* = 0.04) and tumor core (*p* = 0.017), the expression of *SLC27A3* was lower in female patients than in male patients.

The expression of *SLC27A4* was lower in both males and females in the enhancing tumor region (in females *p* = 0.019; in males *p* = 0.004) and tumor core (in males *p* = 0.004) compared to the peritumoral area.

In males, the expression of *SLC27A5* was lower in the glioblastoma tumor (in the enhancing tumor region *p* = 0.05 and in the tumor core *p* = 0.01) compared to the peritumoral area. However, the expression of *SLC27A5* in the tumor core was lower than in the enhancing tumor region (*p* = 0.04). In females, the expression of *SLC27A5* was higher in the tumor core compared to the enhancing tumor region (*p* = 0.019). Additionally, the expression of *SLC27A5* in the tumor core was higher in females than in males (*p* = 0.00003).

The expression of *SLC27A6* was reduced in both sexes (Figure 3). In female patients, the expression of *SLC27A6* in the enhancing tumor region was lower than in the peritumoral area (*p* = 0.04). However, there was no difference in the expression of *SLC27A6* between the tumor core and peritumoral area in female patients (*p* > 0.05). In male patients, the expression of *SLC27A6* in the tumor core was lower than in the peritumoral area (*p* = 0.0009). Additionally, the expression of *SLC27A6* in the tumor core was lower than in the enhancing tumor region (*p* = 0.006).

### 3.3. The Expression of SLC27A1 and SLC27A3 Negatively Correlated with the Expression of SLC27A4, SLC27A5, and SLC27A6

SLC27 representatives may have similar functions [13]. They may be responsible for the channeling of fatty acids for the production of specific lipids [8]. Additionally, the action of individual SLC27 representatives may exclude each other if they induce the channeling of fatty acids to different metabolic pathways. Therefore, to demonstrate the correlations between individual SLC27 representatives, the correlation between the expression levels of the investigated genes was analyzed.

The expression of *SLC27A1* and *SLC27A3* negatively correlated with the expression of *SLC27A4*, *SLC27A5*, and *SLC27A6*. The expression correlation between different regions of the tumor was rare. Only two *SLC27* genes, *SLC27A3* and *SLC27A4*, showed a correlation between their expression in the enhancing tumor region and the tumor core of the glioblastoma. Specifically, a negative correlation was found between the peritumoral area and the tumor core for *SLC27A1*, suggesting differences in fatty acid metabolism between these two regions.

In general, the expression levels of *SLC27* genes were highly correlated within the same group. There was a negative correlation between each *SLC27* gene in the first group and the second group. *SLC27A1* and *SLC27A3* belong to the first group, while *SLC27A4*, *SLC27A5*, and *SLC27A6* belong to the second.

Positive correlations were observed between *SLC27A4* and *SLC27A5*, *SLC27A4* and *SLC27A6*, and *SLC27A5* and *SLC27A6* in all three tumor regions. In addition, a positive correlation was found between the expression of *SLC27A1* and *SLC27A3* in the glioblastoma tumor and between the expression of *SLC27A1* and *SLC27A4* in the tumor core. Negative correlations were observed between *SLC27A3* and *SLC27A4* in the enhancing tumor region and between *SLC27A1* and *SLC27A5* in the tumor core.

In rare cases, there was a correlation between the expression of different *SLC27* genes in different tumor locations (Table 2). For example, a positive correlation was found between *SLC27A3* in the enhancing tumor region and *SLC27A1* in the tumor core, and between *SLC27A4* in the peritumoral area and *SLC27A6* in the tumor core. A negative correlation was found between *SLC27A1* in the peritumoral area and *SLC27A6* in the tumor core, between *SLC27A3* in the enhancing tumor region and *SLC27A5* and *SLC27A6* in the tumor core, and between *SLC27A3* in the tumor core and *SLC27A5* in the peritumoral area.

In women, as in all patients, there was a correlation between the expression of *SLC27A4* with *SLC27A5*, *SLC27A4* with *SLC27A6*, and *SLC27A5* with *SLC27A6* (Table 3). However, no significant correlation was found between the expression of individual *SLC27* genes across different regions of the glioblastoma tumor. Women also showed a negative correlation in the expression of *SLC27A3* in the peritumoral area with the expression of *SLC27A4*, *SLC27A5*, and *SLC27A6* in the enhancing tumor region. Women also showed a negative correlation between the expression of *SLC27A1* in the peritumoral area and the expression of *SLC27A3* in the tumor core.

In men, *SLC27* expression showed correlations similar to the analysis of all patients. There was a correlation of *SLC27A3* expression between the enhancing tumor region and tumor core. Additionally, there was a correlation of *SLC27A6* between the peritumoral area and the tumor core.

Similarly to the analysis of all patients, male samples showed a positive correlation of *SLC27A1* expression with *SLC27A3*, *SLC27A4* with *SLC27A5*, *SLC27A4* with *SLC27A6*, and *SLC27A5* with *SLC27A6* in the studied regions of the glioblastoma tumor (Table 4). Additionally, there was a positive correlation between *SLC27A5* expression in the tumor core and *SLC27A6* expression in the peritumoral area and between *SLC27A5* expression in the peritumoral area and *SLC27A6* expression in the tumor core. There was also a negative correlation between *SLC27A3* expression in the enhancing tumor region and *SLC27A5* in the peritumoral area.

### 3.4. The Expression of SLC27 in the Peritumoral Area Correlated with BMI

The process of carcinogenesis is extremely complex, with some models describing it as a Darwinian process in which the microenvironment plays a significant role [54]. There are various factors that contribute to carcinogenesis, which can be divided into environmental (such as pollution) and patient-related factors. Examples of the latter group include age [55], patient height [56,57], obesity [58], and smoking [59]. These characteristics influence the development of tumors. To investigate the possible correlations between SLC27 and carcinogenesis related to these characteristics, we analyzed the correlation between the expression of the investigated SLC27 genes and patient characteristics such as age, height, weight, BMI, and the number of cigarette packs smoked during their lifetime.

A correlation analysis was conducted on the expression of the investigated *SLC27* representatives with patient characteristics such as age, height, weight, BMI, and the number of cigarette packs smoked in their lifetime. Across all patients, a correlation was found between the expression levels of the investigated *SLC27* and BMI in the peritumoral area. For *SLC27A1*, a positive correlation was observed, while for *SLC27A4*, *SLC27A5*, and *SLC27A6*, a negative correlation was found. Additionally, a negative correlation was found between *SLC27A5* expression in the tumor core and BMI. Furthermore, a negative correlation was found between the expression of *SLC27A1* in the peritumoral area and patient age. Additionally, a positive correlation was observed between the expression of *SLC27A1* in the tumor core and patient weight.

In female patients with glioblastoma, a positive correlation was found between the expression of *SLC27A4*, *SLC27A5*, and *SLC27A6* and the number of cigarette packs smoked in their lifetime. *SLC27A4* and *SLC27A6* also showed a positive correlation in the peritumoral area. Similarly to the analysis of all patients, a negative correlation was observed between the expression of *SLC27A4*, *SLC27A5*, and *SLC27A6* and BMI in the peritumoral area. Furthermore, a negative correlation was found between the expression of *SLC27A1* in the enhancing tumor region and a positive correlation between *SLC27A1* in the enhancing tumor region and patient weight. A negative correlation was also found between the expression of *SLC27A5* and patient weight in the tumor core. Additionally, a negative correlation was observed between the expression of *SLC27A1*, *SLC27A3*, *SLC27A5*, and *SLC27A6* in the peritumoral area and patient age. A correlation was also found between the expression of *SLC27A* and patient age in glioblastoma. Specifically, a positive correlation was found between *SLC27A1* and *SLC27A4* with age in the tumor core, while a negative correlation was found with *SLC27A5*. In the enhancing tumor region, a positive correlation was found between age and the expression of *SLC27A6* in female patients.

In male patients, a correlation was found between the expression of *SLC27* in glioblastoma and BMI (Table 5). A positive correlation was observed between the expression of *SLC27A1* and *SLC27A3* in the enhancing tumor region, while a negative correlation was found with the expression of *SLC27A4*, *SLC27A5*, and *SLC27A6*. In the tumor core, a negative correlation was found between the expression of *SLC27A5* and BMI. Similar to the analyses of all patients, a negative correlation was observed between the expression of *SLC27A5* and BMI in the peritumoral area. Furthermore, a correlation was found between the expression of the investigated *SLC27* and patient age. Specifically, a positive correlation was observed between *SLC27A3* and age in the enhancing tumor region, and a negative correlation was observed with *SLC27A5*.

### 3.5. SLC27 Expression Correlated with Major Fatty Acid Synthesis Enzymes

We conducted an analysis to investigate the correlation between the expression of *SLC27* members and the expression of key fatty acid synthesis enzymes, including *ELOVL1*, *ELOVL3*, *ELOVL6*, *SCD*, and *FADS2*. Expression data for *ELOVL1*, *ELOVL3*, *ELOVL6*, *SCD*, and *FADS2* were obtained from raw data published in our previous papers, where we examined the expression of elongases [47] and desaturases [46] in glioblastomas. *ELOVL6* is involved in the synthesis of stearic acid C18:0 from palmitic acid C16:0 [6], which is responsible for *the* de novo synthesis of the majority of SFA and monounsaturated fatty acid (MUFA). *ELOVL1* and *ELOVL3* are responsible for the further elongation of stearic acid C18:0 up to 26 carbons in length [6]. According to GEPIA, these two elongases play a significant role in the tumorigenesis of glioblastomas. Elevated expression of *ELOVL1* and *ELOVL3* in this tumor type is associated with a poorer prognosis [43]. *SCD* is a desaturase responsible for the production of MUFA from SFA, while *FADS2* is the first enzyme in the biosynthesis pathway of PUFA, regulating the entire pathway [6].

For all patients, we found a positive correlation between the expression of *SLC27A4*, *SLC27A5*, and *SLC27A6,* and the expression of *SCD* and *FADS2* in the peritumoral area. In all studied regions, *SLC27A1* expression was positively correlated with *ELOVL6* expression. Furthermore, *SLC27A3* expression was positively correlated with *ELOVL6* expression in the enhancing tumor region and tumor core. With regard to *ELOVL1* and *ELOVL3*, no significant correlation with the analyzed *SLC27* was found in all patients.

In women, there was a positive correlation between *ELOVL1* expression and *SLC27A*, *SLC27A5*, and *SLC27A6* expression in the tumor core. We observed a positive correlation between the expression of *SLC27A4*, *SLC27A5*, and *SLC27A6* and the expression of *SCD* and *FADS2* in the peritumoral area. Additionally, a positive correlation between the expression of *SLC27A1* and *SCD*, and *FADS2* was observed in the peritumoral area. Finally, a positive correlation between the expression of *SLC27A1* and *ELOVL6* was found in the tumor core, but no correlation was observed in the enhancing tumor region.

In men, a positive correlation was observed between the expression of *ELOVL1* and *SLC27A1*, and *SLC27A3* in the tumor core (Table 6). Furthermore, a negative correlation was found between the expression of *ELOVL1,* and *SLC27A5* and *SLC27A6*, which is opposite to the correlation observed in women. In men, we found a positive correlation between the expression of *SLC27A4* and *SLC27A6* and the expression of *SCD* in the peritumoral area. Additionally, a positive correlation between the expression of *SLC27A1* and *ELOVL6* was observed in all three studied regions.

## 4. Discussion

### 4.1. SLC27 Expression in Glioblastoma

In this study, we showed that *SLC27A4* and *SLC27A6* expression was lower in the glioblastoma compared to the peritumoral area, while the expression of the other examined *SLC27* genes did not differ between the glioblastoma and peritumoral area. However, it is difficult to compare our results with other scientific articles because analyses of *SLC27* expression are not available in the PubMed database (https://pubmed.ncbi.nlm.nih.gov, accessed 25 February 2023). There are only two available studies that examine *SLC27A3* [32,60]. Therefore, to compare our findings with other research groups, we used the Gene Expression Profiling Interactive Analysis (GEPIA) database (http://gepia.cancer-pku.cn, accessed 25 February 2022) [43] and the transcriptomic analysis of different grades of glioma conducted by Seifert et al. [61]. GEPIA contains analyses of gene expression in over 30 types of cancer, including glioblastoma [43]. These are expression analyses taken from The Cancer Genome Atlas (TCGA) [62] and data obtained from healthy tissues from the Genotype-Tissue Expression (GTEx) database [63,64]. Transcriptomic analysis of different grades of glioma conducted by Seifert et al. was analyzed using the REpository of Molecular BRAin Neoplasia DaTa (Rembrandt) [65].

In this study, we found that *SLC27A1* and *SLC27A5* expression did not differ between the glioblastoma and peritumoral area. These results are consistent with GEPIA [43] and Seifert et al. [61]. We also found that there was a lower expression of *SLC27A4* in the glioblastoma compared to the peritumoral area, and these results were consistent with Seifert et al. [61]. However, GEPIA reports that *SLC27A4* expression does not differ between glioblastomas and healthy brain tissue [43]. We also found lower expression of *SLC27A6* in the glioblastoma compared to the peritumoral area. This is in contrast to GEPIA [43] and Seifert et al. [61], where *SLC27A6* expression did not differ between the glioblastoma and healthy brain tissue.

In this study, we did not obtain results for *SLC27A2*, an enzyme characterized by very low expression in the brain [21,29,30]. GEPIA analyses also show that the level of *SLC27A2* expression is about 200 times lower in glioblastoma than among other *SLC27* genes [43]. As a result, it can be inferred that the level of *SLC27A2* expression is below the detection threshold of the method used in this study. Furthermore, the low expression of *SLC27A2* in glioblastoma tumors suggests that it does not play a significant role in the tumorigenesis processes in glioblastoma. This is supported by the lack of association between the level of *SLC27A2* expression in glioblastoma tumors and patient prognosis [43].

### 4.2. Differences in SLC27 Expression in Glioblastoma Tumors between Sexes

In glioblastoma tumors, specifically in the tumor core, women exhibit lower expression of *SLC27A1* and *SLC27A3*, but higher expression of *SLC27A5* than men. This suggests differences in fatty acid metabolism between sexes. However, there is a lack of research on sex differences in *SLC27* expression in the brain or even in other organs, and therefore, we cannot refer to the existing literature.

SLC27 shows similar activity to specific fatty acids [13]. However, SLC27A3 and SLC27A5 demonstrate weaker fatty acid activation properties compared to other SLC27 [13]. Moreover, SLC27A3, SLC27A5, and SLC27A6 show weak or no uptake properties of fatty acids [13]. SLC27A1 participates in the transport of fatty acids across the BBB [22], indicating that in women, the uptake of fatty acids from the bloodstream by tumor core cells in glioblastoma may be less intense than in men.

SLC27A3 participates in ceramide synthesis, as demonstrated in experiments on U-87 MG cells [32]. This suggests that the production of ceramides in glioblastoma tumors may be less intense in women. Ceramides play a critical role in the tumorigenesis processes in glioblastoma [32,66,67] and the higher expression of SLC27A3 is associated with a worse prognosis [32]. Therefore, this may explain the better prognosis for women with glioblastoma [68].

### 4.3. Correlations between the Expression of Various SLC27

Analyses of the correlation between different *SLC27* in various studied regions indicate certain dependencies in fatty acid metabolism. A positive correlation of *SCL27A4* expression was found between the tumor core and enhancing tumor region but not with the peritumoral area. A negative correlation of *SLC27A1* expression was observed between the tumor core and the peritumoral area. *SCL27A1* and *SCL27A4* are important in fatty acid uptake through BBB [22]. This suggests that fatty acid uptake through BBB is similar in the two studied regions of the glioblastoma tumor, but differs significantly from the peritumoral area. Among the other *SLC27*, a positive correlation of *SLC27A3* expression was found between all studied regions of glioblastoma. SLC27A3 is responsible for the production of ceramides, as demonstrated in experiments on U-87 MG cells [32]. This correlation indicates that ceramide synthesis may be similar between tumors and the rest of the brain. However, the lack of correlation of other *SLC27* suggests that lipid metabolism significantly differs between the tumor and peritumoral area.

In our study, we found a negative correlation between the expression of *SLC27A3* and the expression of *SLC27A4* and *SLC27A5*, and a positive correlation between the expression of *SLC27A3* and *SLC27A1*. SLC27A3 is a protein that does not participate in fatty acid uptake [13]. It is responsible for channeling fatty acids to produce ceramides, as demonstrated in experiments on U-87 MG cells [32]. SLC27A1 is responsible for the synthesis of triglycerides [16,17,18], ceramides [9], and β-oxidation [18,19], as demonstrated in other models. Meanwhile, SLC27A4 is responsible for the production of triglycerides [17,36], cholesterol esters [36], and ceramide [9]. SCL27A1 and SCL27A4 are important in fatty acid uptake through BBB [22]. SLC27A5 may be responsible for the production of triglycerides, glycerophospholipids, and cholesteryl esters [39]. The observed correlations suggest that ceramide biosynthesis may occur in glioblastoma, which precludes the use of fatty acids for the production of other lipids. Ceramide production is also positively correlated with the expression and uptake of fatty acids by SLC27A1. Moreover, a positive correlation for expression was found between *SLC27A4*, *SLC27A5*, and *SLC27A6*, while a negative correlation was observed between *SLC27A1* and *SLC27A5* expression. This suggests that the production of triglycerides, glycerophospholipids, and cholesteryl esters may depend on the uptake of fatty acids through BBB, depending on SLC27A4 but not SLC27A1.

In the tumor core, but not in the enhancing tumor region, there was a positive correlation between the expression of *SLC27A1* and *SLC27A4*, both of which are involved in fatty acid uptake through the BBB [22]. This suggests that fatty acid uptake in the tumor core is mediated by these two proteins, whose expression levels are positively correlated. In contrast, there was no correlation between the expression of *SLC27A1* and *SLC27A4* in the enhancing tumor region, suggesting that fatty acid uptake in this region may depend on one of these proteins.

### 4.4. Correlation of SLC27 Expression with Patient Characteristics

The correlation of *SLC27* gene expression with patient characteristics was analyzed, including age, BMI, height, and smoking history. Negative correlations were observed between the expression of *SLC27A4*, *SLC27A5*, and *SLC27A6* and BMI in the peritumoral area of the patients, with *SLC27A5* and *SLC27A6* showing this correlation in both genders. Additionally, the expression of *SLC27A1* was positively correlated with BMI in all patients. SLC27A1 and SLC27A4 are involved in the uptake of fatty acids by the BBB [22], with SLC27A1 specifically participating in the transport of DHA C22:6n-3 [23,24]. This suggests that in obese patients, there may be a lower expression of *SLC27A4* and higher expression of *SLC27A1* in the brain, indicating greater uptake of DHA C22:6n-3 by the brain. The results also indicate that lipid metabolism in the brain may be altered in obese individuals, but further research is needed to fully understand the significance of SLC27 in obesity. Currently, no experimental studies are available on this topic.

In women, there was a negative correlation between the expression of *SLC27A1* and weight and BMI in the enhancing tumor region and a positive correlation with weight in the tumor core. This suggests differences in the uptake and metabolism of fatty acids between these two regions of glioblastoma tumors in lean and obese women. In the peritumoral area, there was also a negative correlation of *SLC27* with weight in women. According to one available study [69], obese women have lower expression of *SLC27A1* in muscle tissue compared to lean women, which may explain the negative correlation between *SLC27A1* expression and BMI and weight in the enhancing tumor region of glioblastoma tumors and the peritumoral area in women. However, the cited study [69] shows no difference in *SLC27A1* expression between obese and lean men. In this study, a positive correlation was found between BMI and *SLC27A1* expression in the enhancing tumor region of glioblastoma tumors in men, which is inconsistent with the cited study that examined expression in muscle tissue. In men with glioblastoma tumors, there was also a negative correlation between the expression of *SLC27A4*, *SLC27A5*, *SLC27A6*, and a positive correlation between *SLC27A1* and *SLC27A3* expression with BMI. This indicates significant differences in fatty acid metabolism between obese and lean men, with greater uptake of fatty acids by the BBB involving SLC27A1 and less involving SLC27A4 in obese men [22]. This may result in increased uptake of DHA C22:6n-3 by glioblastoma tumors in obese men [23,24]. Obese men may also have higher production of ceramides [32] and less intensive metabolism of other types of lipids [39] compared to lean men.

In women, there was a negative correlation between the expression of the tested *SLC27* genes and age in the peritumoral area, whereas no such relationship was observed in men. These results may indicate changes in fatty acid metabolism in the healthy brain that occur with age, especially in women. However, there is a lack of experimental studies in this direction. In the tumor core of women, there was a negative correlation between the expression of *SLC27A5* and a positive correlation between the expression of *SLC27A4* and *SLC27A1* with age. Additionally, in the enhancing tumor region, there was a positive correlation between the expression of *SLC27A6* and age. In men, there was a negative correlation between the expression of *SLC27A5* and a positive correlation between the expression of *SLC27A3* with age in the enhancing tumor region. These findings also indicate that the patient’s age and sex significantly affect fatty acid metabolism in glioblastoma tumors.

In women, there was a positive correlation between the expression of *SLC27A4*, *SLC27A5*, and *SLC27A6* with cigarette smoking in all investigated regions of the glioblastoma tumor and in the peritumoral area, whereas no such relationship was observed in men. These data suggest that cigarette smoking in women with glioblastoma may affect carcinogenesis and, consequently, may influence fatty acid metabolism in the tumor. However, there is a lack of available studies on the influence of cigarette smoking on the expression of *SLC27* enzymes in tumors, including lung cancer. Therefore, further research is needed to investigate the association between cigarette smoking and fatty acid metabolism in tumors in women.

### 4.5. Correlation with Enzymes Involved in Fatty Acid Synthesis

We have demonstrated a positive correlation between the expression of *SLC27A1* and *SLC27A3* and the expression of *ELOVL6* in both the glioblastoma tumor and the peritumoral area. Furthermore, we have identified a positive correlation between the expression of *SLC27A4*, *SLC27A5*, and *SLC27A6* and the expression of *SCD* and *FADS2* in the peritumoral area. These findings shed some light on the metabolic pathways of fatty acids in glioblastoma tumors.

ELOVL6 participates in the synthesis of stearic acid C18:0 from palmitic acid C16:0 [6], while SCD and FADS2 are involved in the production of MUFAs and PUFAs, respectively. SLC27 proteins transport fatty acids across the cell membrane and activate the transported fatty acids [13,28,35]. These proteins exhibit activity towards various fatty acids from different groups [13]. Specifically, they activate SFAs (palmitic acid C16:0, stearic acid C18:0, lignoceric acid C24:0, and cerotic acid C26:0 [12,13,37]), MUFAs (oleic acid C18:1n-9) [13], and PUFAs (arachidonic acid C20:4n-6) [13]. However, the possibility of transporting and activating other fatty acids (such as linoleic acid C18:2n-6) has not been studied. The high activity of SLC27 towards various fatty acids suggests that other fatty acids may also be transported and activated by these proteins.

SLC27A1 and SLC27A4 are the most important SLC27 proteins involved in the transport of fatty acids across the BBB [22]. Our data suggest a correlation between the transport of palmitic acid C16:0 by SLC27A1 across the BBB and the elongation of this fatty acid. This provides insights into the potential effects of drugs targeting SLC27A1. SLC27A3 exhibits similar properties, although it does not account for fatty acid uptake [13]. It may be crucial for intracellular fatty acid metabolism, particularly in the synthesis of ceramides, as evidenced by experiments on U-87 MG cells [32]. Moreover, fatty acids transported by SLC27A4 across the BBB undergo different modifications in the peritumoral area. SFAs (palmitic acid C16:0 and stearic acid C18:0) are desaturated, i.e., transformed into MUFAs. In contrast, 18-carbon PUFAs are activated and transformed into longer PUFAs, such as arachidonic acid C20:4n-6, eicosapentaenoic acid C20:5n-3, and docosahexaenoic acid C22:6n-3. Interestingly, the observed correlations indicate that SLC27A5 and SLC27A6 exhibit similar activities to SLC27A4, although a correlation with *FADS2* was only observed in the peritumoral area. This suggests that such a correlation does not occur in the glioblastoma tumor itself.

In women, a positive correlation was observed between the expression of *ELOVL1* and *SLC27A4*, *SLC27A5*, and *SLC27A6* in the tumor core. This finding suggests that very-long-chain SFAs may be taken up from the blood by SLC27A4 and subsequently elongated by ELOVL1 to be incorporated into triglycerides, glycerophospholipids, and cholesteryl esters in the tumor core. In contrast, in men, a positive correlation was found between the expression of *ELOVL1* and *SLC27A1* and *SLC27A3*, and a negative correlation with *SLC27A5* and *SLC27A6*, which is opposite to the correlation observed in women. This indicates that the metabolism of very-long-chain SFAs differs between sexes. In men, very-long-chain SFAs may be elongated and incorporated into ceramides after uptake by SLC27A1.

### 4.6. Clinical Significance of the Obtained Results

Analyses on the GEPIA portal show that the expression level of any *SLC27* gene is not significantly associated with prognosis for patients with glioblastoma [43]. However, comparing the highest and lowest quartiles of *SLC27* expression, a trend towards worse outcomes with higher expression of *SLC27A4* (*p* = 0.062) and *SLC27A3* (*p* = 0.083) was observed for glioblastoma patients [43]. This suggests that these two SLC27 representatives may have important clinical significance as a therapeutic target.

According to Kolar et al., analyzing The Chinese Glioma Genome Atlas, higher expression of *SLC27A3* in glioblastoma tumors is associated with worse outcomes [32]. This suggests that this protein plays a significant role in the tumorigenic processes in this cancer type. This enzyme activates fatty acids and channels them towards ceramide production [32]. SLC27A3 is essential in the stemness and self-renewal of glioblastoma stem cells [33], as well as in the proliferation and anchorage-independent growth of glioma cells [60]. Currently, no published studies on the significance of SLC27A4 in glioblastoma tumorigenesis are available on the PubMed browser (https://pubmed.ncbi.nlm.nih.gov/, accessed 25 February 2022). However, SLC27A4 is known to play a crucial role in brain physiology by participating in the transport of fatty acids across the BBB, together with SLC27A1 [22]. This suggests that SLC27A4 may also be important in the uptake of fatty acids from the bloodstream by glioblastoma cells. Nonetheless, SLC27A4 may also be significant in the tumorigenesis of other cancers. Higher expression of *SLC27A4* in breast tumors is associated with worse prognosis, particularly with distant metastasis-free survival [70]. In breast cancer cells, SLC27A4 plays a role in proliferation and epithelial-to-mesenchymal transition (EMT) [70]. Similarly, studies on hepatocellular carcinoma (HCC) cells have demonstrated that SLC27A4 is crucial for the proliferation and invasion of these cells [71]. In lung cancer cells, SLC27A4 is responsible for chemoresistance [72].

The results of this study have shown that the expression of *SLC27A3* was higher in the glioblastoma tumors of men than of women. Additionally, the expression of *SLC27A3* in the enhancing tumor region of men was positively correlated with the age and BMI of the patient. Furthermore, the expression of *SLC27A3* in men was positively correlated with the expression of *ELOVL1* in this region of the glioblastoma tumor. *ELOVL1* is an enzyme responsible for the elongation of SFA [6], and the expression of this enzyme is also associated with poor patient outcomes in glioblastoma [43]. These observations may be useful for the development of personalized therapies, especially for obese men who are much older than the average glioblastoma patient. Such therapies should target the enzymes responsible for the synthesis of ceramides containing very-long-chain SFAs (either the studied SLC27A3 or ELOVL1) or enzymes directly responsible for ceramide synthesis.

Another enzyme described in this study that may affect outcomes in glioblastoma patients is *SLC27A4*. We have shown that the expression of *SLC27A4* was the same in both sexes. It was negatively correlated with BMI and height in men, and positively correlated with age and the intensity of cigarette smoking in women. In women, but not in men, the expression of *SLC27A4* was positively correlated with the expression of *ELOVL1*, indicating a different function of this enzyme between sexes. Drugs targeting SLC27A4 could be helpful in treating glioblastoma in low-weight men who are lean or thin. In female glioblastoma patients, such drugs could have a therapeutic effect in old age, and in cases when the glioblastoma developed as a result of intense cigarette smoking.

### 4.7. Limitations

The objective of this study was to investigate the expression of SLC27 in glioblastoma tumors and to explore the correlation between the expression of the investigated proteins and the key enzymes involved in fatty acid biosynthesis, as well as patient characteristics such as sex, age, BMI, and smoking. However, there are several limitations to the study. The expression levels of the investigated genes in glioblastoma tumors were compared to those in the peritumoral area rather than healthy brain tissue due to ethical issues with obtaining samples of healthy brain tissue. While epilepsy surgery brain samples are sometimes used as a control, they may be more similar to brain tumors than healthy brain tissue [50]. Therefore, using the peritumoral area as a control seems to be the most reasonable option. Another limitation is the small sample size of patients, all of whom come from a single population. Nevertheless, the obtained results are consistent with the available literature data, suggesting that the conclusions drawn from the study may provide valuable data for scientific research. Additionally, the study did not investigate the correlation between the investigated genes and the expression of lipid biosynthesis enzymes such as sphingolipids, glycerophospholipids, and triacylglycerols. As a result, the significance of SLC27 in the entire lipid metabolism of glioblastoma tumors was not fully demonstrated, indicating a potential direction for further research into lipid metabolism in glioblastomas.

## 5. Conclusions

Several conclusions can be drawn from the results of this study. The expression of SLC27, responsible for transporting fatty acids through the BBB, is reduced in glioblastoma tumors. This suggests a lower uptake of fatty acids by the tumor compared to non-tumor brain tissue. Therefore, it can be inferred that glioblastoma tumors exhibit a more intense biosynthesis of fatty acids than their uptake from the bloodstream. This metabolic difference might set them apart from healthy brain tissue. Thus, a therapeutic approach aimed at de novo synthesis of fatty acids may be more effective than targeting the transport of fatty acids.

Furthermore, SLC27 is associated with the de novo synthesis of fatty acids. Specifically, SLC27A1 and SLC27A3 are associated with stearic acid synthesis, and SLC27A4, SLC27A5, and SLCA6 with very-long-chain-fatty-acid synthesis. There are differences in the expression of these genes between genders, which showed a positive correlation in women and negative correlation in men.

Lastly, the transport and activation of fatty acids in glioblastoma tumors may depend on the intensity of cigarette smoking in women and BMI in men. This factor may influence the development of a personalized therapy approach for each patient.

## Figures and Tables

**Figure 1 brainsci-13-00771-f001:**
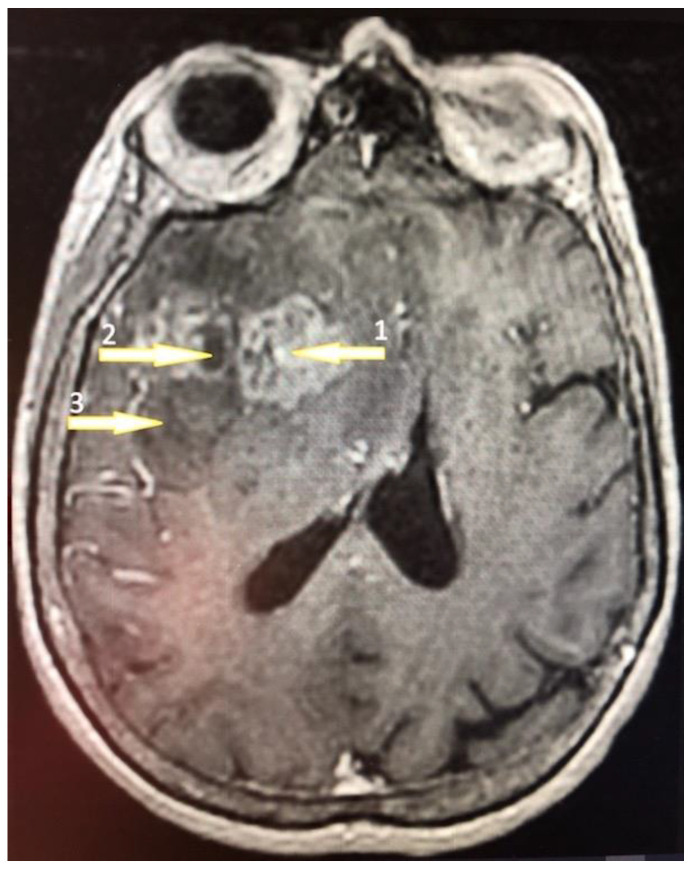
A 61-year-old high school teacher was admitted to the department of neurosurgery on a day of a seizure episode. She was confused and developed a palmomental reflex on her right side with no other symptoms. Brain nuclear magnetic resonance (NMR) with contrast enhancement revealed a multiform expansive mass in the left frontal lobe with an irregularly shaped growth zone (1) and necrotic core (2) surrounded by the peripheral edematic area (3). The tumor was removed totally by craniotomy and the patient was discharged on the 8th day postoperatively. The pathology revealed a GBM diagnosis and the patient was treated subsequently by an irradation dose of 60 Gy.

**Figure 2 brainsci-13-00771-f002:**
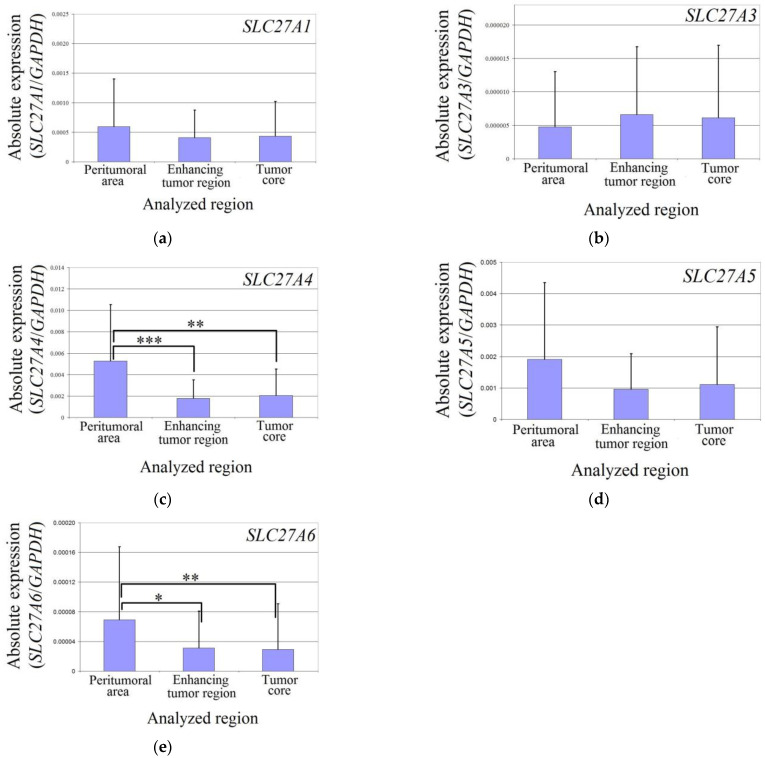
Expression of *SLC27As* in glioblastoma tumor and peritumoral area. The expression level of *SLC27A1* (**a**), *SLC27A3* (**b**), *SLC27A4* (**c**), *SLC27A5* (**d**), and *SLC27A6* (**e**) in the peritumoral area, enhancing tumor region, and tumor core. *—denotes statistically significant differences in the expression of the gene between different regions of a glioblastoma tumor as determined by the Wilcoxon signed-rank test (*p* < 0.05); **—*p* < 0.01; ***—*p* < 0.001.

**Figure 3 brainsci-13-00771-f003:**
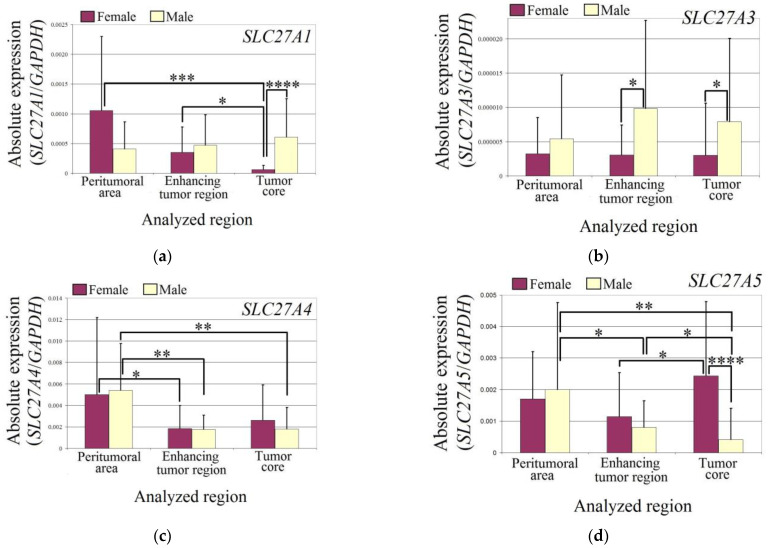
The expression levels of *SLC27A* genes (**a**) *SLC27A1* (**b**) *SLC27A3* (**c**) *SLC27A4* (**d**) *SLC27A5* (**e**) *SLC27A6* in the glioblastoma tumor and peritumoral area in male and female patients. Wilcoxon signed-rank tests were used to estimate statistically significant differences in gene expression (*p* < 0.05 for *, *p* < 0.01 for **, *p* < 0.001 for ***, and *p* < 0.0001 for ****).

**Table 1 brainsci-13-00771-t001:** Characteristics of the patients.

	Mean	Standard Deviation	Minimum	Maximum	Mean	Standard Deviation	Minimum	Maximum	Mean	Standard Deviation	Minimum	Maximum
	All Patients				Women				Men			
Age at surgery	60.7	12.5	36	81	60.8	14.1	36	79	60.6	11.9	41	81
Weight	84	19	55	130	70.3	15	55	95	93.8	15.4	73	130
Height	172	12	147	196	162	8	147	173	178	8.6	163	196
BMI	28.7	4.8	21.5	38.9	27.4	5.7	21.5	36.2	29.5	4.2	24.7	38.9
Pack years	13.4	14.0	0	42	5.6	9.2	0	21	18.6	14.5	0	42

**Table 2 brainsci-13-00771-t002:** The correlation between the expression of different SLC27 genes in different tumor regions in all patients.

Studied Gene/Location		*SLC27A1*			*SLC27A3*			*SLC27A4*			*SLC27A5*			*SLC27A6*		
		PA	ETR	TC	PA	ETR	TC	PA	ETR	TC	PA	ETR	TC	PA	ETR	TC
*SLC27A1*	PA	1.00														
	ETR	0.09	1.00													
	TC	−0.43 *	0.13	1.00												
*SLC27A3*	PA	0.21	0.15	−0.10	1.00											
	ETR	0.21	0.46 *	0.48 *	0.46 *	1.00										
	TC	−0.21	0.29	0.77 *	0.45 *	0.72 *	1.00									
*SLC27A4*	PA	−0.20	0.06	0.00	−0.06	−0.05	−0.19	1.00								
	ETR	−0.43	−0.08	0.09	−0.37	−0.49 *	−0.23	0.11	1.00							
	TC	−0.20	0.03	0.44 *	−0.21	−0.14	0.05	0.18	0.58 *	1.00						
*SLC27A5*	PA	−0.16	0.09	−0.19	−0.42 *	−0.21	−0.46 *	0.69 *	0.14	0.01	1.00					
	ETR	−0.37	−0.26	−0.08	−0.37	−0.61 *	−0.32	0.08	0.86 *	0.32	0.09	1.00				
	TC	0.13	−0.18	−0.54 *	−0.05	−0.56 *	−0.57 *	0.32	0.15	0.40 *	0.40	0.11	1.00			
*SLC27A6*	PA	−0.26	0.09	−0.11	−0.17	−0.21	−0.32	0.71 *	0.34	0.13	0.77 *	0.14	0.35	1.00		
	ETR	−0.28	−0.22	−0.02	−0.13	−0.34	−0.12	0.10	0.69 *	0.24	0.11	0.84 *	−0.05	0.12	1.00	
	TC	−0.46 *	−0.35	−0.06	−0.10	−0.49 *	−0.13	0.59 *	0.41	0.60 *	0.34	0.30	0.65 *	0.40	0.32	1.00

The values of Spearman’s rank correlation coefficients are given. * statistically significant correlation of the expression of two genes/locations (*p* < 0.05). PA—peritumoral area; ETR—enhancing tumor region; TC—tumor core.

**Table 3 brainsci-13-00771-t003:** The correlations between the expression of different *SLC27* genes in different tumor regions in all female patients.

Studied Gene/Location		*SLC27A1*			*SLC27A3*			*SLC27A4*			*SLC27A5*			*SLC27A6*		
		PA	ETR	TC	PA	ETR	TC	PA	ETR	TC	PA	ETR	TC	PA	ETR	TC
*SLC27A1*	PA	1.00														
	ETR	−0.20	1.00													
	TC	−0.59	−0.45	1.00												
*SLC27A3*	PA	0.15	−0.31	−0.31	1.00											
	ETR	−0.07	0.12	−0.15	0.55	1.00										
	TC	−0.76 *	0.12	0.47	0.48	0.41	1.00									
*SLC27A4*	PA	0.48	0.50	−0.41	−0.26	0.10	−0.22	1.00								
	ETR	−0.14	0.11	−0.03	−0.86 *	−0.54	−0.52	0.38	1.00							
	TC	−0.37	−0.11	0.20	−0.64	−0.67	−0.35	0.03	0.69	1.00						
*SLC27A5*	PA	0.12	0.49	−0.07	−0.26	0.29	−0.44	0.60 *	0.48	−0.09	1.00					
	ETR	−0.07	−0.07	0.02	−0.84 *	−0.73 *	−0.40	0.19	0.87 *	0.67	−0.12	1.00				
	TC	0.37	0.13	−0.54	0.18	−0.24	−0.36	0.82 *	0.36	0.72 *	0.61	0.12	1.00			
*SLC27A6*	PA	0.17	0.36	−0.44	−0.04	0.51	−0.44	0.45	0.36	−0.66	0.85 *	−0.07	0.50	1.00		
	ETR	0.02	−0.31	0.22	−0.71 *	−0.58	−0.43	0.26	0.78 *	0.64	0.07	0.92 *	0.05	0.10	1.00	
	TC	−0.32	−0.19	0.20	−0.13	−0.61	0.11	0.34	0.65	0.96 *	−0.13	0.62	0.77 *	−0.45	0.62	1.00

The values of Spearman’s rank correlation coefficients are given. * statistically significant correlation of the expression of two genes/locations (*p* < 0.05). PA—peritumoral area; ETR—enhancing tumor region; TC—tumor core.

**Table 4 brainsci-13-00771-t004:** The correlation between the expression of different *SLC27* genes in different tumor regions in all male patients.

Studied Gene/Location		*SLC27A1*			*SLC27A3*			*SLC27A4*			*SLC27A5*			*SLC27A6*		
		PA	ETR	TC	PA	ETR	TC	PA	ETR	TC	PA	ETR	TC	PA	ETR	TC
*SLC27A1*	PA	1.00														
	ETR	0.42	1.00													
	TC	0.18	0.19	1.00												
*SLC27A3*	PA	0.12	0.26	−0.11	1.00											
	ETR	0.18	0.67 *	0.37	0.32	1.00										
	TC	−0.15	0.41	0.52	0.38	0.65 *	1.00									
*SLC27A4*	PA	−0.35	0.01	−0.19	0.15	−0.30	−0.13	1.00								
	ETR	−0.46	−0.30	0.09	−0.09	−0.42	0.19	0.26	1.00							
	TC	0.18	0.16	0.42	−0.36	0.13	0.03	0.24	0.26	1.00						
*SLC27A5*	PA	−0.36	−0.04	−0.45	−0.44	−0.70 *	−0.47	0.70 *	0.35	0.04	1.00					
	ETR	−0.46	−0.39	−0.19	0.09	−0.58	−0.17	0.17	0.92 *	−0.04	0.40	1.00				
	TC	−0.16	0.04	−0.37	−0.01	−0.20	−0.43	0.63 *	−0.26	0.45	0.55	−0.56	1.00			
*SLC27A6*	PA	−0.30	−0.12	−0.33	−0.49	−0.59	−0.53	0.67 *	0.46	0.32	0.93 *	0.35	0.71 *	1.00		
	ETR	−0.44	−0.17	−0.56	−0.20	−0.48	−0.24	0.24	0.76 *	−0.48	0.45	0.88 *	−0.39	0.29	1.00	
	TC	−0.55	−0.56	−0.50	−0.11	−0.33	−0.27	0.70 *	0.08	0.20	0.70 *	0.01	0.81 *	0.80 *	0.12	1.00

The values of Spearman’s rank correlation coefficients are given. * statistically significant correlation of the expression of two genes/locations (*p* < 0.05). PA—peritumoral area; ETR—enhancing tumor region; TC—tumor core.

**Table 5 brainsci-13-00771-t005:** Correlation between expression of analyzed *SLC27* genes and expression of key enzymes involved in the fatty acid synthesis.

Studied Gene	Location	Age at Surgery	Weight	Height	BMI	Pack Years	Age at Surgery	Weight	Height	BMI	Pack Years	Age at Surgery	Weight	Height	BMI	Pack Years
		All Patient					Women					Men				
SLC27A1	PA	−0.46 *	0.10	0.04	0.43 *	0.14	−0.68 *	−0.47 *	0.26	0.06	−0.11	0.15	0.05	−0.04	0.28	−0.02
	ETR	−0.06	−0.09	0.29	−0.37	0.15	−0.20	−0.38 *	0.26	−0.82 *	−0.11	0.03	0.62 *	0.26	0.61 *	0.40
	TC	0.23	0.55 *	0.32	0.29	0.30	0.74 *	0.68 *	0.20	0.34	−0.29	−0.19	−0.15	0.37	0.22	−0.02
*SLC27A3*	PA	−0.12	0.00	0.05	0.26	0.19	−0.61 *	−0.15	0.24	0.16	−0.10	0.18	0.05	−0.22	0.27	0.12
	ETR	−0.04	0.29	0.16	0.29	0.44	−0.31	−0.32	−0.09	−0.19	0.18	0.59 *	0.31	−0.38	0.70 *	0.41
	TC	0.12	0.34	0.18	0.23	0.14	0.25	0.35	0.24	−0.10	−0.08	0.17	0.06	−0.03	0.26	0.13
SLC27A4	PA	−0.12	−0.35	0.03	−0.48 *	0.06	−0.32	−0.76 *	−0.13	−0.71 *	0.70 *	−0.08	0.05	0.37	0.01	−0.26
	ETR	−0.01	0.08	0.14	−0.37	0.28	0.27	0.04	−0.18	−0.32	0.64 *	0.01	−0.51 *	−0.17	−0.42 *	−0.40
	TC	0.14	0.14	0.14	0.04	0.13	0.43 *	0.19	−0.39	0.21	0.94 *	−0.37	0.32	−0.63 *	−0.05	−0.26
*SLC27A5*	PA	−0.19	−0.44 *	0.00	−0.57 *	−0.03	−0.44 *	−0.92 *	−0.36	−0.75 *	0.33	−0.07	−0.41 *	0.04	−0.52 *	−0.15
	ETR	0.15	−0.05	−0.08	−0.34	0.18	0.37	0.20	−0.13	−0.07	0.52 *	−0.44 *	−0.12	0.44 *	−0.61 *	−0.15
	TC	−0.09	−0.57 *	−0.24	−0.37 *	−0.27	−0.70 *	−0.46 *	0.14	−0.39	0.84 *	0.21	−0.25	0.22	−0.37 *	0.03
*SLC27A6*	PA	−0.30	−0.20	0.35 *	−0.48 *	−0.02	−0.42 *	−0.94 *	−0.31	−0.86 *	0.64 *	−0.32	−0.16	0.42 *	−0.36	−0.30
	ETR	0.16	0.12	0.05	−0.19	0.44	0.46 *	0.20	−0.18	−0.04	0.76 *	−0.07	−0.34	−0.02	−0.66 *	0.12
	TC	0.08	−0.02	0.07	−0.22	−0.10	0.21	0.21	0.05	−0.10	0.89 *	−0.06	−0.03	0.31	−0.30	−0.37

Spearman’s rank correlation coefficients are given. *—statistically significant correlation of the expression of *SLC27* gene with patient characteristics (*p* < 0.05). PA—peritumoral area; ETR—enhancing tumor region; TC—tumor core.

**Table 6 brainsci-13-00771-t006:** Correlation between the studied *SLC27* genes and the expression of key enzymes involved in fatty acid synthesis.

Studied Gene	Location	*ELOVL1*	*ELOVL3*	ELOVL6	*SCD*	*FADS2*	*ELOVL1*	*ELOVL3*	*ELOVL6*	*SCD*	*FADS2*	*ELOVL1*	*ELOVL3*	*ELOVL6*	*SCD*	*FADS2*
		All Patient					Women					Men				
*SLC27A1*	PA	0.12	−0.25	0.48 *	0.34	0.20	−0.71	0.23	0.33	0.74 *	0.67 *	0.34	−0.32	0.54 *	0.15	−0.06
	ETR	0.13	−0.34	0.40 *	0.29	0.26	0.29	−0.19	−0.01	0.41	−0.03	−0.01	−0.29	0.89 *	−0.05	0.35
	TC	0.17	−0.02	0.66 *	0.10	0.03	0.09	0.20	0.66 *	−0.04	−0.02	0.49 *	−0.12	0.57 *	0.07	−0.11
*SLC27A3*	PA	0.05	−0.09	0.25	0.10	0.27	−0.26	−0.02	0.41	0.25	0.29	−0.09	−0.14	0.19	0.00	0.14
	ETR	−0.18	−0.15	0.44 *	−0.03	0.10	−0.14	0.11	0.37	−0.07	−0.27	−0.06	−0.26	0.46	−0.18	0.38
	TC	0.14	−0.05	0.44 *	0.28	0.19	−0.40	0.01	0.29	0.25	−0.32	0.54 *	−0.06	0.35	0.05	0.28
*SLC27A4*	PA	−0.06	−0.07	−0.03	0.53 *	0.39 *	0.30	0.23	−0.17	0.74 *	0.69 *	−0.25	−0.16	0.03	0.49 *	0.27
	ETR	0.38	−0.12	−0.18	0.30	0.17	0.47	−0.28	−0.45	0.31	0.52	0.22	0.16	0.05	0.12	−0.33
	TC	0.13	−0.10	0.15	−0.12	0.11	0.68 *	−0.02	−0.09	−0.33	0.45	−0.22	−0.29	0.36	0.14	−0.08
*SLC27A5*	PA	−0.11	−0.14	−0.31	0.45 *	0.41 *	0.47	−0.17	−0.21	0.69 *	0.70 *	−0.28	−0.11	−0.32	0.34	0.34
	ETR	0.20	−0.06	−0.31	0.30	0.17	0.14	−0.37	−0.52	0.25	0.47	0.16	0.24	−0.22	0.22	−0.28
	TC	−0.12	−0.04	−0.22	−0.30	−0.23	0.66 *	0.24	0.50	−0.35	−0.45	−0.53 *	−0.31	−0.03	−0.04	−0.43
*SLC27A6*	PA	0.13	−0.24	−0.29	0.50 *	0.42 *	0.63	−0.29	−0.17	0.66 *	0.72 *	−0.12	−0.23	−0.34	0.48 *	0.36
	ETR	0.00	−0.12	−0.09	0.16	0.10	0.05	−0.37	−0.23	0.03	0.21	−0.13	0.05	0.02	0.22	−0.01
	TC	−0.10	−0.03	−0.16	−0.07	0.03	0.71 *	0.18	−0.15	−0.13	0.42	−0.59 *	−0.19	−0.10	0.15	−0.14

Spearman’s rank correlation coefficients are given. *—statistically significant correlation (*p* < 0.05). PA—peritumoral area; ETR—enhancing tumor region; TC—tumor core.

## Data Availability

The data presented in this study are available on request from the corresponding author.

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
