# Peer review of "Reduced Expression of Very-Long-Chain Acyl-CoA Synthetases SLC27A4 and SLC27A6 in the Glioblastoma Tumor Compared to the Peritumoral Area"

_brainsci, 2023, doi:10.3390/brainsci13050771_

Round 1

Reviewer 1 Report

Comments and Suggestions for Authors

Authors investigate (perturbed) SLC27 signaling components in GBM patients.
Limitations such as the lack of comparison to non-malign vs peritumor tissue could be discussed.
It would be possible to collect data from other literature search tools as scholar, DEGs in HF2303 GBM Cells on Sox2 Knockdown would be available f.i., or genecards information, giving more insight into signaling. Some references that could be included in the background as https://doi.org/10.1146/annurev-physiol-032122-030352
Side note: Typo famale in figures.

Continued success with your ongoing research!

Author Response

Rev 1.

Authors investigate (perturbed) SLC27 signaling components in GBM patients.
Limitations such as the lack of comparison to non-malign vs peritumor tissue could be discussed.

Due to ethical concerns, it is not possible to obtain healthy brain tissue for research purposes. Therefore, scientists commonly rely on analyses of the peritumoral brain zone as a viable alternative. In our work, we drew upon the study conducted by Lemée et al., which compared the peritumoral brain zone of glioblastomas with brain samples obtained during epilepsy surgery. The authors concluded that the peritumoral brain zone of glioblastomas is a suitable control material, as discussed in the limitations section. This approach has become a widely accepted standard in brain tumor research.

Lemée, J.M.; Com, E.; Clavreul, A.; Avril, T.; Quillien, V.; de Tayrac, M.; Pineau, C.; Menei, P. Proteomic analysis of glioblastomas: What is the best brain control sample? J Proteomics 2013, 85, 165–173.

It would be possible to collect data from other literature search tools as scholar, DEGs in HF2303 GBM Cells on Sox2 Knockdown would be available f.i., or genecards information, giving more insight into signaling. Some references that could be included in the background as https://doi.org/10.1146/annurev-physiol-032122-030352

A paragraph about signal transduction induced by fatty acids has been added.

Side note: Typo famale in figures.

The typo has been corrected.

Continued success with your ongoing research!

Thank you very much.

Reviewer 2 Report

Comments and Suggestions for Authors

the Authors present the results of a very complex study investigating the expression levels of SLC27 in GBM, as well as the correlation between expression levels and patients characteristics.

I can offer the following comments:

- in the abstract, please clarify the importance of evaluating SCL27 expression in GBM. Not all readers are familiar with this.

- the Introduction is quite long and hard to follow. I understand that it is crucial to provide background, but I encourage the Authors to make an effort to shorten the Introduction and make it more direct to the point.

- Information on the enrolled patients should be expanded and maybe summarized in a table.

- Please expand on the clinical relevance of the results. At present, I am not sure that I can fully understand it.

Comments on the Quality of English Language

Some polishing is required, but I undestand that English is not Authors' first language (nor mine).

Author Response

Rev 2.

the Authors present the results of a very complex study investigating the expression levels of SLC27 in GBM, as well as the correlation between expression levels and patients characteristics. I can offer the following comments:

- in the abstract, please clarify the importance of evaluating SCL27 expression in GBM. Not all readers are familiar with this.
The abstract has been modified.

- the Introduction is quite long and hard to follow. I understand that it is crucial to provide background, but I encourage the Authors to make an effort to shorten the Introduction and make it more direct to the point.
The Introduction has been abridged.

- Information on the enrolled patients should be expanded and maybe summarized in a table.
We have added a table that includes the information on patiens.

- Please expand on the clinical relevance of the results. At present, I am not sure that I can fully understand it.
The clinical significance of the obtained results has been expanded on in accordance with the reviewer's recommendation.

Some polishing is required, but I undestand that English is not Authors' first language (nor mine).
The article has been checked again by a native speaker.

Reviewer 3 Report

Comments and Suggestions for Authors

The authors  aimed to analyze solute carrier family 27 (SLC27) in glioblastoma tumors. These enzymes are responsible for transporting fatty acids across the cell membrane and converting them into fatty acyl-CoA. Tumor samples were collected from a total of 28 patients and analyzed using quantitative real-time polymerase chain reaction (qRT-PCR) to determine the expression levels of SLC27.

The study also sought to explore the relationship between SLC27 expression and patient characteristics, such as age, height, weight, body mass index (BMI), and smoking history, as well as the expression levels of enzymes responsible for de novo fatty acid synthesis.

Among the authors’ findings:

-The expression of SLC27A4 and SLC27A6 was significantly lower in glioblastoma tumors compared to the peritumoral area.

-Moreover, women exhibited lower expression of SLC27A4 in the tumor, whereas men had lower expression of SLC27A5.

-Notably, a positive correlation was observed between the expression of SLC27A4, SLC27A5, and SLC27A6 and smoking history in women, whereas men exhibited a negative correlation between these SLC27 and BMI. -The expression of SLC27A1 and SLC27A3 was positively correlated with the expression of ELOVL6.

The authors concluded that their  findings suggest that the metabolism of fatty acids differs between glioblastoma tumors and healthy brain tissue.

Interesting study.

I have some minor suggestions.

-Abstract must be improved better summarizing the sections

-purpose is “Hence, the current study aimed to scrutinize the role of SLC27 proteins, which are 149

responsible for the uptake, activation, and channeling of fatty acids”. Please expend it. Use bullet points if you need.

-Introduce the results themes using a few sentences before

-Add the labels to the figures presenting data

-Insert the limitations in the discussions

-Tables do not follow the MDPI standard

Author Response

Rev 3.

I have some minor suggestions.

-Abstract must be improved better summarizing the sections.
The abstract has been modified.

-purpose is “Hence, the current study aimed to scrutinize the role of SLC27 proteins, which are 149

The sentence has been changed.

-responsible for the uptake, activation, and channeling of fatty acids”. Please expend it. Use bullet points if you need.
The passage has been changed.

-Introduce the results themes using a few sentences before
The description of results has been modified.

-Add the labels to the figures presenting data
The labeling of the X-axis has been added.

-Insert the limitations in the discussions
The limitations of this study are now presented in the discussion.

-Tables do not follow the MDPI standard
The formatting of the tables has been changed.

Round 2

Reviewer 2 Report

Comments and Suggestions for Authors

thanks for having considered my comments